# System Design and Modeling of a High Temperature PEM Fuel Cell Operated with Ammonia as a Fuel

**Giovanni Cinti [1],\* , Vincenzo Liso [2] , Simon Lennart Sahlin [2] and Samuel Simon Araya [2]**

[1]   Department of Engineering, University of Perugia, Via G. Duranti 93, 06125 Perugia, Italy
[2]   Department of Energy Technology, Aalborg University, Pontoppidanstræde 111, 9220 Aalborg, Denmark;
    vli@et.aau.dk (V.L.); sls@et.aau.dk (S.L.S.); ssa@et.aau.dk (S.S.A.)
\*   Correspondence: giovanni.cinti@unipg.it

**Abstract:** Ammonia is a hydrogen-rich compound that can play an important role in the storage of green hydrogen and the deployment of fuel cell technologies. Nowadays used as a fertilizer, $NH_3$ has the right peculiarities to be a successful sustainable fuel for the future of the energy sector. This study presents, for the first time in literature, an integration study of ammonia as a hydrogen carrier and a high temperature polymer electrolyte membrane fuel cell (HT-PEMFC) as an energy conversion device. A system design is presented, that integrates a reactor for the decomposition of ammonia with an HT-PEMFC, where hydrogen produced from $NH_3$ is electrochemically converted into electricity and heat. The overall system based on the two technologies is designed integrating all balance of plant components. A zero-dimensional model was implemented to evaluate system efficiency and study the effects of parametric variations. Thermal equilibrium of the decomposition reactor was studied, and two different strategies were implemented in the model to guarantee thermal energy balance inside the system. The results show that the designed system can operate with an efficiency of 40.1% based on ammonia lower heating value (LHV) at the fuel cell operating point of 0.35 A/cm$^2$ and 0.60 V.

**Keywords:** ammonia; high temperature PEM; fuel cell

## 1. Introduction

Due to its versatility, hydrogen has been recently gaining traction as energy storage solution. Even though it can be produced from a variety of energy sources, its most attractive feature is the possibility of producing it from renewable energy resources and using it in various applications, either directly or after converting it into other chemical products both for energy consumption and further chemical processing.

Hydrogen is converted into electrical energy with high efficiencies in fuel cells. However, at ambient conditions (25 °C and 1 bar), hydrogen has low density of only 0.0813 g L$^{-1}$, which requires either a high pressure storage, e.g., 700 bar for automotive application, which increases the density to 40 g L$^{-1}$ and the corresponding volumetric energy density of 5.6 MJ L$^{-1}$, or liquid state storage, for many practical applications [1,2]. In both cases, hydrogen undergoes thermodynamic transformations, which can increase the overall storage and transportation costs significantly. Hydrogen can also be stored in carbon nanotubes, metallic hydrides, or complex hydrides under more moderate temperature and pressure conditions, but with only limited gravimetric density [2,3].

Another solution is to store and transport hydrogen indirectly in the chemical bonds of other chemicals, thereby increasing volumetric energy density and, consequently, transportability. Methanol and ethanol are some of the most common examples of such hydrogen rich alcohols, where hydrogen is chemically bonded with carbon, also known as carbon-based fuels [4–6]. An issue

that makes carbon-based fuels less appealing is the involvement of carbon dioxide, both in the synthesis process and in the power conversion. $CO_2$ has to be supplied to the chemical plant and released during the power production process. This means that two different cycles have to be closed: the hydrogen cycle and the carbon dioxide cycle. While hydrogen can be made entirely renewable by relying on renewable energy sources, such as wind and solar for the hydrogen production via water electrolysis, for the carbon cycle to close, the carbon dioxide should either come from carbon-capture or from biomass sources [4].

An interesting alternative to this is the carbonless molecule of ammonia. In the case of ammonia, the gas involved is nitrogen, which means that the additional loop that has to be closed is the $N_2$ cycle. Nitrogen can be separated from air to feed the chemical process, usually the Haber–Bosh process, and released back to the atmosphere during energy consumption, making the overall system completely carbon-free [7]. Recently, alternative green ammonia synthesis methods, including electrochemical synthesis at lower temperature and pressure conditions, are being investigated [7–9].

Therefore, one of the advantages of the ammonia over other carbon-based fuels is the absence of $CO_2$ emissions at the point of use. For carbon-based fuels on the other hand, even when the carbon cycle is closed, $CO_2$ is still emitted locally and the power generation is not carbon-free. However, concerning local emissions, if $NH_3$ is used in combustion engines, the productions of $NO_x$ should be considered [7], an issue that is significantly alleviated or even eliminated when using $NH_3$ in fuel cell systems. Moreover, the availability of nitrogen is much higher than carbon dioxide. If, for instance, atmospheric air is considered as a source for both gases, nitrogen is available at high concentrations of around 79%, while carbon dioxide is in the range of hundreds of ppm (416 ppm as of June 2020 [10]). However, it is worth mentioning that some sites, including biogas plants, bioethanol plants, and emission-intensive industries, can be used as sources of high concentration of $CO_2$ to produce carbon-based fuels, such as methanol through the power-to-X scheme [4]. The concentration of $N_2$ and $CO_2$ in the feedstock is directly related not only to the separation cost, both in terms of energy and economics, but also to the availability of the quantities necessary to feed the respective fuel synthesis plants.

$NH_3$ has a high hydrogen concentration of 75 mol% and is liquid at a relatively low pressure of 10 bar, with high energy density of 15.6 MJ $L^{-1}$ compared to liquid hydrogen, which has an energy density of 9.1 MJ $L^{-1}$ at cryogenic temperature or compressed hydrogen, 5.6 MJ $L^{-1}$ at 70 MPa [11]. Moreover, ammonia as a well-known fertilizer with a mature production technology, is one of the most produced chemicals worldwide, which can count on a well-established distribution network [11]. However, its application in the energy sector is not significant and has only been investigated more recently as a fuel in several combustion-based energy systems. Experiences are reported in power systems based on gas turbine technology and on internal combustion engines [7,11]. In general, combustion of ammonia suffers from low flame speed and high resistance to auto-ignition, which calls for pre-mixing with other fuels as combustion promoters, e.g., hydrogen, that can be obtained with a partial decomposition of ammonia itself [11].

An alternative solution is the use of ammonia in fuel cells [12]. Fuel cells are electrochemical devices that directly transform the chemical energy of fuels into electricity without the typical emissions of combustion-based cycles. Ammonia can be coupled with fuel cells directly in solid oxide fuel cells (SOFC) and direct ammonia fuel cells (DAFC) or via a pre-decomposition into nitrogen and hydrogen in polymer electrolyte membrane fuel cells (PEMFC) [13,14]. Due to the peculiarity of fuel cells, the presence of nitrogen in the fuel stream only acts as a diluent of hydrogen and does neither generate any pollutant nor cause significant performance decay, allowing all fuel cells to operate with a mixture of nitrogen and hydrogen.

Even though it is possible to design a system were ammonia is decomposed and then the fuel stream is fed to any fuel cell unit, such a coupling should consider the presence of ammonia in the decomposed gas, which may poison the membrane electrode assembly (MEA) of certain fuel cell types. For instance, PEMFCs are extremely sensitive to ammonia contamination, where $NH_3$ poisons the

Pt/C anode catalyst and reacts with the acidic Nafion membrane [15,16]. According to ISO14687-2, the maximum level of concentration of ammonia in hydrogen used in the PEM fuel cell vehicle is 0.1 ppm [13]. To reach the ISO target, it is necessary to introduce a purification technology after ammonia decomposition [13]. A concept design of a PEMFC-based system is proposed in [17], where the decomposition reactor is heated electrically, reaching 99.5% conversion of ammonia. However, to the knowledge of the authors, only one experience that couples the ammonia decomposition reactor and PEM fuel cells has been reported, where 500–1000 ppm of ammonia impurity from the ammonia cracking chamber is eliminated in a selective ammonia oxidation (SAO) reactor before the produced hydrogen can be used in a PEMFC [18]. The introduction of an additional gas purification phase increases the cost of the system and introduces additional energy losses. In general, the lower the tolerance of the end use device to ammonia, the higher the cost of the clean-up.

Another promising fuel cell technology is the high temperature PEM fuel cell (HT-PEMFC). In an HT-PEMFC, the traditional Nafion membrane, used in the low temperature PEM fuel cell technology, is substituted with a phosphoric acid-doped polybenzimidazole (PBI) membrane that operates in the temperature range 120–200 °C [19]. The higher operating temperatures increase the tolerance to impurities, for example, CO tolerance increases up to 3% compared to only few ppm in low temperature PEM fuel cells [19]. Due to this higher tolerance to impurities, HT-PEMFCs have been extensively studied in conjunction with methanol reformer, where the effects of the different resulting impurities, namely CO, $CO_2$, and $CH_3OH$ have been reported with satisfactory tolerances for single cells and stacks [20–22]. Moreover, steam methane reforming and HT-PEMFC systems are also available in the literature [23–25].

However, despite the popularity of ammonia as a chemical commodity, and its potential as a storage for renewable hydrogen, its use in HT-PEMFCs has been largely ignored. While it is expected that traces of $NH_3$ will react with the PA-based electrolyte, the tolerance of an HT-PEMFC to $NH_3$ is only reported in [26], where a general tolerance to percentage of $NH_3$ is reported from an internal report. The development of HT-PEMFC technology and the strong interest in the use of ammonia as a fuel opens the possibility for the development of ammonia-fed HT-PEM fuel cell systems.

Therefore, this study presents for the first time in literature, a system study of an HT-PEMFC stack coupled with the ammonia decomposition unit. An integrated system design has been developed and thermodynamic studies of the system model were performed. A complete ammonia decomposition process is assumed, and hence, the corresponding high ammonia cracking temperatures, where full ammonia conversion is expected to take place, are used in the model. Consequently, the experiments were also done only for the dilution effect of nitrogen in the feed-gas and ammonia slip was ignored.

## 2. Materials and Methods

In this section, the modeling approach for the ammonia-fuel cell system is described. The fuel cell electrical performance obtained by the experimental test is used as input for the novel heat integrated system concept of an ammonia-fueled HT-PEMFC. The system model was studied with zero-dimensional thermodynamic models of the reactors and heat exchangers. The model was implemented in a calculation sheet (MSExcel©) that calculates energy balance and gas flows. The gas thermo-physical properties were taken from a freely available database (NIST-JANAF [27]). The reaction mechanisms for the ammonia decomposition and the ammonia and hydrogen combustion process were implemented in Cantera [28] using a Python programming language script.

### 2.1. Experimental

Experiments for the current work were carried out in a Greenlight Innovation fuel cell test station on a 37 cell HT-PEMFC stack with an active area of 165 $cm^2$. The tests were first done with pure hydrogen for 24 h at 0.4 A $cm^{-2}$ and the stoichiometric ratios were set to 1.3 on the anode side and 2.5 on the cathode side. Successively, a test with gas mixture containing 68.3% hydrogen and 31.7% nitrogen in the anode feed was performed for the same duration at the same conditions of current

density and stoichiometric ratios. A typical HT-PEMFC operating temperature of 160 °C was chosen for the tests. Galvanostatic polarization measurements were used for characterizing the different operating conditions. For this, the current was varied from 0 to 75 A at smaller steps of 2.5 A in the range between 0–10 A to better capture the activation overpotential and a current step of 5 A was used for the remainder of the polarization measurements.

## 2.2. Modeling

A schematic of the system design is shown in Figure 1. The design of the new system is based on the experience obtained from reformed methanol-fed HT-PEMFC systems. In a methanol-based system, the fuel is first reformed into a hydrogen rich gas mixture that contains $CO_2$, traces of CO, and unconverted methanol before entering the fuel cell unit. The heat required for the reforming process is supplied by a burner, which is first fed with a separate flow of methanol dedicated to the combustion reaction during the startup phase and then successively sustained by the anode off-gas [4]. In the current work, a similar approach was considered, where ammonia is first decomposed, and the resulting hydrogen–nitrogen mixture is fed to the fuel cell. The main challenge, compared to the methanol-based system, is the higher operating temperature of the ammonia decomposition reactor, which ranges between 550–900 °C [13,29], compared to the relatively low methanol steam reforming reactor of around 200–300 °C [4]. These higher operating temperatures require an optimized and innovative energy balance strategy. Moreover, the decomposition temperature is strongly related to the ammonia tolerance level of the HT-PEMFC. The lower the tolerance, the higher the decomposition temperature. However, it is worth mentioning that more recent advances in materials have shown that near equilibrium conversion of ammonia can be achieved at temperatures below 500 °C with higher activity catalysts based on ruthenium and alkali-based catalysts such as sodium and lithium imide [29].

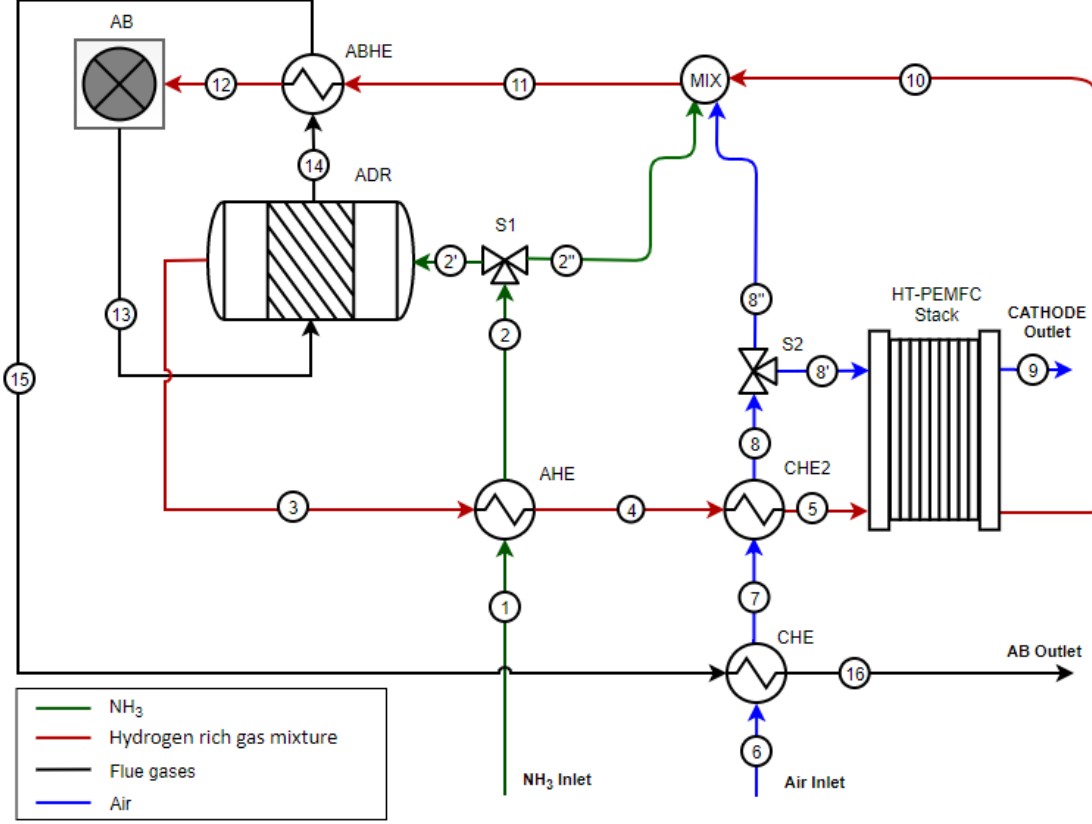

**Figure 1.** Conceptual schematic of the integrated ammonia decomposition reactor-HT-PEMFC system.

In the current work, a complete decomposition of ammonia is considered for the system design, even though, it has been reported that the HT-PEMFC can tolerate some ammonia in the anode feed [26], allowing for an even simpler system design. However, further studies are required to better understand the effects of ammonia on the performance and durability of HT-PEMFCs before a system can be designed around the assumption of tolerance to traces of ammonia.

Ammonia decomposition is an endothermic reaction and requires external heat to maintain the desired operating temperature. Analogous to a methanol reformer, in the proposed scheme, the heat for ammonia decomposition is supplied by an after burner (AB). In the schematic shown in Figure 1, the gas flow pipes are numbered from 1 to 16. Since ammonia enters the system at ambient temperature, it is pre-heated before entering the ammonia decomposition reactor (ADR) in the ammonia heat exchanger (AHE). Before entering the reactor, a splitter allows to separate part of the ammonia and send it via the mixer to the afterburner. However, ammonia–air mixture suffers from low flame rate and high resistance to auto-ignition, leading to longer ignition delay times, and therefore, it is normally mixed with other fuels for combustion [11,30]. In the current system, the splitter can be activated in specific operating conditions to support combustion, for example, during the start-up or when the unit operates in off design conditions. However, at nominal operating conditions, the design assumes that the entire flow of ammonia is directed to the ADR, which is modeled as an equilibrium reactor with a heat exchanger. The heat in this case is supplied by the anode off gasses of the fuel cell, which are combusted in the afterburner.

The decomposed ammonia is then cooled down to the fuel cell temperature in the AHE, where heat is recovered and used to pre-heat the inlet ammonia to 30 °C below the ADR operating temperature to guarantee the heat recovery. Before reaching the fuel cell inlet, the decomposed mixture enters a second heat exchanger, called cathodic heat exchanger 2 (CHE2), where the remaining heat is used to pre-heat the inlet cathodic air up to the fuel cell temperature.

The oxygen in the air is necessary for both the HT-PEMFC and for the afterburner. In the splitter S2, the total inlet air is separated in two different streams: the cathodic gas flow (8′ in the schematic), and the afterburner air flow (8″ in the figure). The cathodic off gasses (pipe 9 in the figure) are vented into the atmosphere, while the anodic gas flow that contains the unreacted hydrogen is sent to the afterburner to complete oxidation and provide heat for the system thermal integration. The anode off gasses (pipe 10), the afterburner air flow (pipe 8″), and in some specific operating conditions that require more heat the ammonia flow (pipe 7″), are mixed in the mixer. A high temperature heat exchanger, afterburner heat exchanger (ABHE), pre-heats the gas mixture before reaching the afterburner. The level of pre-heat is defined by the outlet temperature of the gas mixture ($T_{12}$ in the schematic). This temperature is a design parameter for the system and is discussed in the results section. The afterburner exhausts are cooled in the three components ADR, ABHE, and CHE before they are vented into the atmosphere (gas stream 16 in the figure).

The design allows to optimize the heat recovery and to guarantee the operating conditions of the base components: HT-PEMFC and ADR. All heat exchangers, including the ADR, are modeled with the following energy balance equation:

$$\Delta h_{hot} = \eta_{HE} \Delta h_{cold},\tag{1}$$

where $\Delta h$ is the difference between outlet and inlet gas enthalpy of heat exchanger of hot gasses ($\Delta h_{hot}$) and cold gasses ($\Delta h_{cold}$) and $\eta_{HE}$ is the heat exchanger efficiency.

Both the afterburner and the mixer are modeled as adiabatic units with no thermal losses. In the afterburner, the oxidation of the fuel is completed, and the produced heat increases the off gasses temperature. The afterburner is designed based on the excess of oxygen parameter ($EO_{AB}$), which is the molar ratio between the inlet oxygen and the stoichiometric value. This parameter allows to calculate the air flow for the afterburner and to design the splitter S2.

The HT-PEMFC is designed based on the experimental results for the electrochemical performances and on the gas thermos-physical properties for the energy balance. HT-PEMFC operating conditions are

defined by operating temperature ($T_{Cell}$), current density (J), and gas flowrates. The gas flowrates into the two electrodes of the fuel cell depend on the anode stoichiometry ($\lambda_{anode}$) and cathode stoichiometry ($\lambda_{cathode}$) set points and are used to calculate the ammonia flowrate into the ADR. The fuel cell is modeled as the isothermal unit and a specific heat flow $\dot{Q}_{FC}$ is calculated from the energy balance and has to be subtracted by the fuel cell cooling system. The energy balance of the HT-PEMFC is calculated as follows:

$$\Delta h_{anode} + \Delta h_{cathode} + P_{FC} + \dot{Q}_{FC} = 0, \tag{2}$$

where $P_{FC}$ is the specific power density of the HT-PEMFC, $\Delta h_{anode}$ and $\Delta h_{cathode}$ are the specific enthalpy difference between outlet and inlet gasses of anode and cathode, respectively. The parameters that are necessary to calculate the system energy balance are reported in Table 1, where all the parameters (flow rates, current, power, etc.) are per unit of the fuel cell active area.

**Table 1.** Model input parameters.

| Inlet Parameter | Symbol | Unit |
|---|---|---|
| Anode Stoichiometry | $\lambda_{anode}$ | - |
| Cathode Stoichiometry | $\lambda_{cathode}$ | - |
| Current density | J | A cm$^{-2}$ |
| Cell temperature | $T_{cell}$ | °C |
| Excess Oxygen after burner | $EO_{AB}$ | - |
| AB inlet temperature | $T_{12}$ | °C |
| ADR Temperature | $T_{ADR}$ | °C |
| Heat Exchanger efficiency | $\eta_{HE}$ | - |

The model allows to calculate the energy balance of all components, gas composition, temperature, and flow rate of all gas flows. At the fuel cell level, it is possible to calculate operating voltage, power density ($P_{FC}$), and heat flow ($\Delta H_{FC}$). At the system level, efficiency $\eta_s$ is calculated as follows:

$$\eta_s = \frac{P_{FC}}{m_{NH_3} \cdot LHV_{NH_3}}, \tag{3}$$

where $m_{NH_3}$ is the ammonia specific flow rate and $LHV_{NH_3}$ is the ammonia lower heating value.

## 3. Results

In this section, the results produced from the modeling of the ammonia decomposition process and the $NH_3/H_2$/air combustion are presented. The experimental results on the performance of the fuel cell stack in the presence of nitrogen in the anode feed gas have also been presented. These models serve to set the temperature limits for the operation of the ADR and AB in the system model. Furthermore, the overall system performance at different operating conditions is described.

### 3.1. Complete Ammonia Decomposition and Its Effects on the Fuel Cell Stack Perfomance

Hydrogen is produced via ammonia decomposition through the following endothermic reaction ($\Delta H = 45.6$ kJ/molNH$_3$) without the addition of oxygen or steam:

$$NH_3 \rightleftharpoons \frac{3}{2}H_2 + \frac{1}{2}N_2. \tag{4}$$

Ganley et al. [31] reported that the reaction must be enabled by a catalyst. The kinetics of the reaction have been investigated with several supported metals as catalysts (Ru, Ir, Ni, Rh, Pt, Pd, Fe), where the highest catalytic activity is observed over Ru-based catalysts and a comparable high catalytic activity has been obtained at higher temperatures on the less expensive Ni-based catalysts [32]. In an experimental study, Chellappa et al. [33] found that ammonia decomposition

can be considered a first order reaction, with the reaction rate depending solely on the ammonia concentration. Differently from the ammonia synthesis, the decomposition can be conducted at low pressures (1–2 atm) for thermodynamic reasons (principle of Le Chatelier). The ammonia conversion in the decomposition reactor can be calculated as:

$$X_{NH_3} = 1 - \frac{X_{NH_3,out}}{X_{NH_3,in}} \, ,$$ (5)

where $X_{NH_3,in}$ and $X_{NH_3,out}$ are the ammonia molar fractions at the inlet and outlet of the reactor. The factor indicates how much ammonia is converted into hydrogen. From the experimental and numerical analysis in [34], we can see in Figure 2 that above a temperature of 600 °C, ammonia can be fully converted in the reactor over a wide range of ammonia flow rates, which in the figure, is expressed using the factor W/F i.e., Ni–Pt/Al$_2$O$_3$ catalyst loading weight to ammonia flow rate ratio. This is consistent with findings in [32] where ammonia decomposition was tested on nickel- and ruthenium-based catalysts. Besides, we can see that to achieve the same ammonia conversion values, higher ammonia flow rates will require higher temperatures of operation.

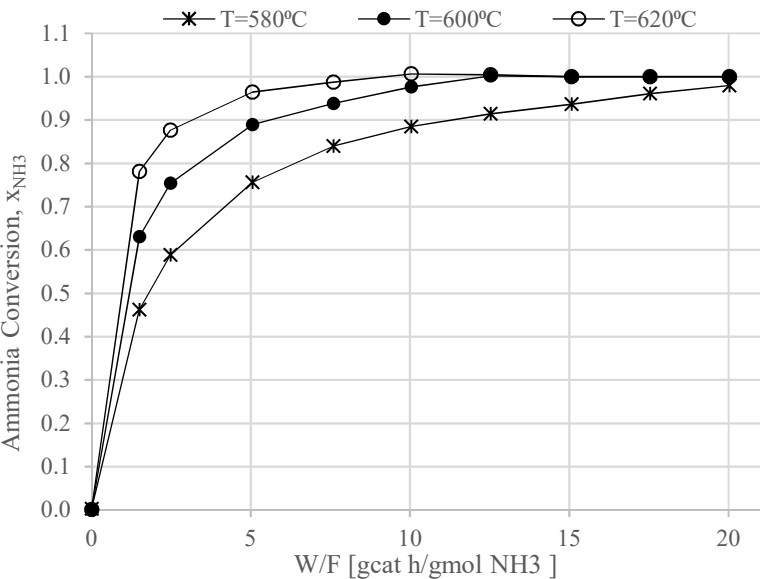

**Figure 2.** Effect of temperature on ammonia decomposition at different contact time i.e., catalyst weight/volume flow rates (W/F) using Ni–Pt/Al$_2$O$_3$ catalyst (Adapted from [34] with permission from Elsevier).

The theoretical limits of the ammonia conversion can be deduced from the ammonia decomposition thermodynamic equilibrium composition at atmospheric pressure at different temperatures in Figure 3. It can be seen that higher temperatures produce higher ammonia conversions.

The experimental data shown in Figure 4 show that 31.7% nitrogen in the anode stream has only slight dilution effect on the fuel cell stack performance. This value is above the maximum concentration of nitrogen that can be obtained from the complete decomposition of ammonia (Figure 3), and hence, it can be concluded that in the absence of NH$_3$ slip in the feed gas (via complete decomposition or gas purification), a hydrogen–nitrogen gas mixture at concentrations corresponding to the typical ammonia decomposition process does not degrade the fuel cell performance.

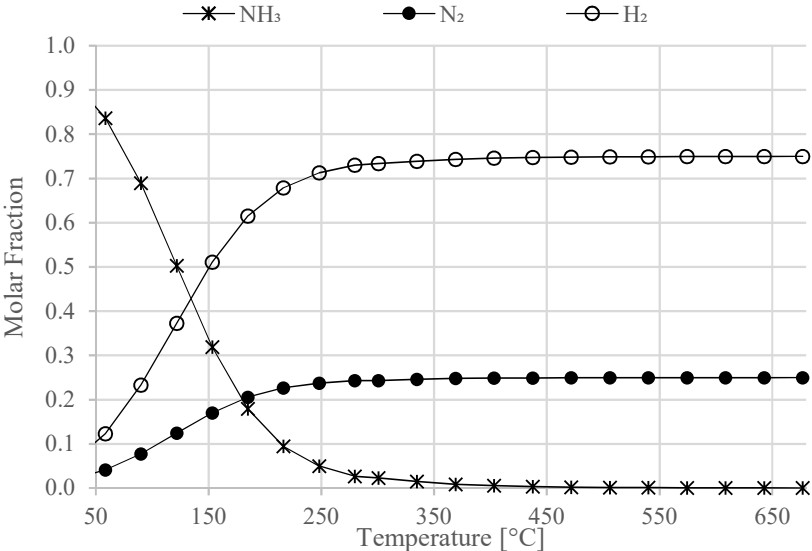

**Figure 3.** Molar fraction composition of ammonia decomposition at atmospheric pressure based on chemical equilibrium.

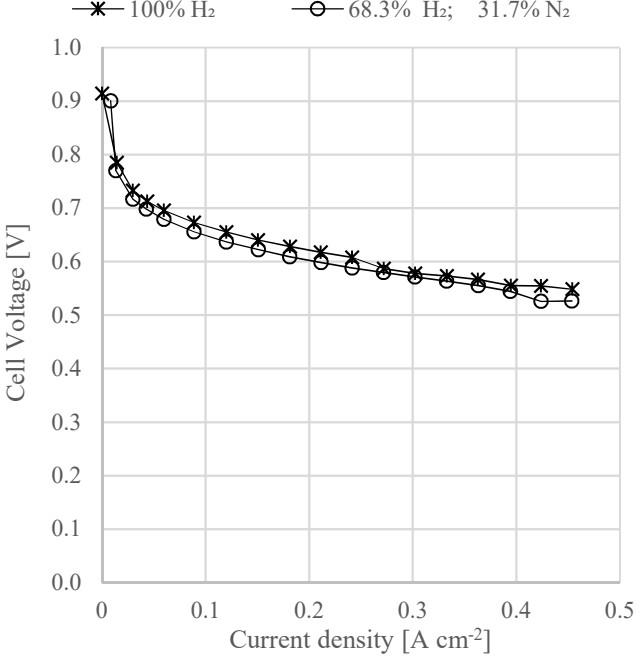

**Figure 4.** Experimental polarization curves of a HT-PEMFC stack fueled by hydrogen and a mixture of hydrogen and nitrogen.

### 3.1.1. Afterburner

In the afterburner reactor, hydrogen and ammonia are combusted to produce the heat required for the endothermic ammonia decomposition reaction. Ammonia and hydrogen combustion has previously been studied, where it has been shown that ammonia combustion in air is difficult due to the high auto ignition temperature and low laminar flame velocity [35–37]. However, it has been demonstrated that the swirling flow of the burnt mixture can facilitate the process by providing hot gases and radicals to the incoming fresh fuel and by increasing the residence time [38]. Ammonia combustion is especially necessary during the system start-up when the ammonia decomposition has not reached the operating temperature and hydrogen is not yet available in the stream at the burner inlet.

Chemical kinetics of $NH_3/H_2$/air combustion is complex—detailed kinetics models have been studied, among others, by Otomo et al. [36] and by Nakamura et al. [39]. In this study, the reaction mechanism in [39] has been implemented to estimate the NOx concentration in the burnt gases at different temperatures. The reaction mechanism was implemented in Cantera [28] using a Python programming language script. The afterburner is simulated as a "continuous stirred-tank reactor" (CSTR) at a pressure of 1 atm. The simulated inlet gas compositions consider the stoichiometric combustion of $NH_3$/air for the system start-up and the $H_2$/air combustion at different operating values of the anode stoichiometry and the excess air in the afterburner.

In air pollution, parts per million (ppm) by mole or by volume is used to account for the amount of pollutants. These quantities can be assumed equal in the case of an ideal gas. From Figure 5, it can be seen that the $NO_x$ concentration in ppm is affected by combustion temperature in the afterburner. The model results show that while NO production in the case of the ammonia/air combustion starts to increase continuously already at temperatures below 1200 °C, their production from $H_2$/air combustion remains negligible below 1370 °C. Therefore, since the burner operates under $H_2$/air combustion at nominal conditions, the latter temperature has been chosen as the upper limit for the adiabatic combustion temperature in the system modeling. Furthermore, a local peak in NO concentration is visible in the temperature range of the ammonia autoignition of around 650 °C. In [40], the NO formation pathways from ammonia combustion were described and it was shown that in the ignition region, the majority of atomic nitrogen is oxidized to NO.

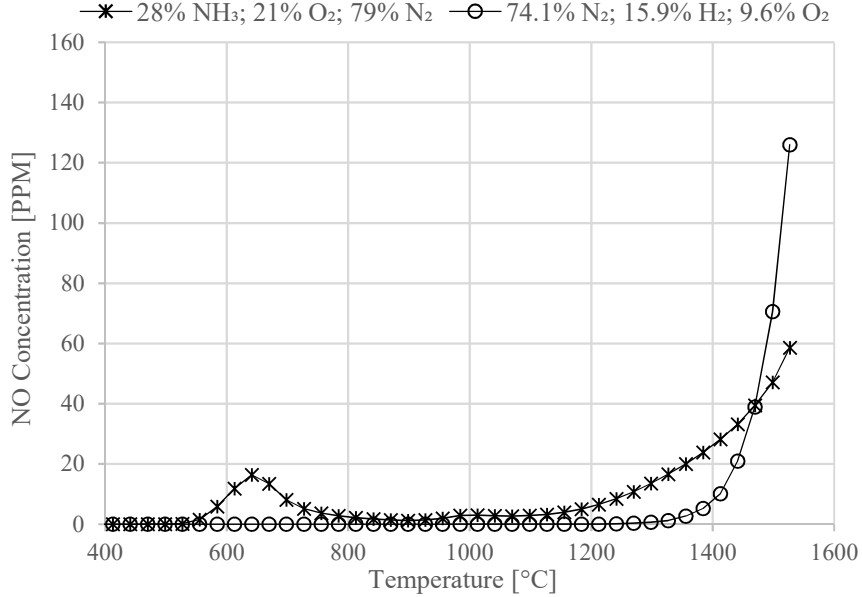

**Figure 5.** Concentration of NO in ppm in the afterburner outlet gases due to $NH_3/H_2$/air combustion. One case refers to the combustion of ammonia during the system startup, while the other curve refers to the $H_2$/air combustion.

### 3.1.2. System Modeling

The model implemented in the calculation sheet was studied by varying the operating parameters. The selected nominal conditions are reported in Table 2.

**Table 2.** Nominal operating parameters of the system.

| Parameter | Value |
|-----------|-------|
| $\lambda_{anode}$ | 1.37 |
| $\lambda_{cathode}$ | 2.5 |
| J | 0.35 A cm$^{-2}$ |
| $T_{cell}$ | 160 °C |
| $EO_{AB}$ | 1.65 |
| $T_{AB,in}$ | 330 °C |
| $T_{ADR}$ | 600 °C |
| $\eta_{HE}$ | 0.85 |

Anode stoichiometry was set to 1.37, whereby the excess amount of hydrogen that does not react in the fuel cell is oxidized in the afterburner to provide the heat necessary to balance the internal heat requirement of the system, in particular, for the ammonia decomposition reactor. Note that this value is slightly higher than the one used in the experimental test and voltage values calculated by the model may slightly underestimate real values. However, in an experimental work performed in a similar short HT-PEMFC stack, it was found that once hydrogen starvation is avoided at around an anode stoichiometric ratio of 1.3, further increase in $\lambda_{anode}$ does not improve the fuel cell stack performance [41].

Cathode stoichiometry was selected from the experimental data to 2.5 and is a standard operating condition for an HT-PEMFC. Typical HT-PEMFC operating conditions were also used for current density and temperature, at 0.35 a cm$^{-2}$ and 160 °C, respectively.

The ammonia decomposition reaction temperature was set to 600 °C, which, according to the literature data in Figure 2, gives complete ammonia decomposition at satisfactory *W/F* ratio. In spite of the fact that complete ammonia conversion can be achieved at the chosen operating temperature, in reality, ammonia concentration in the ppb level may be present in the decomposed mixture. However, this limit is compatible with the HT-PEMFC tolerance available in literature [26]. To keep the afterburner temperature below the risk of NOx formation, the afterburner inlet temperature, $T_{12}$, was set to 330 °C. The effect of this parameter on the system energy balance is discussed below. All heat exchangers operate with an efficiency of 0.85.

This optimization follows the afterburner excess of oxygen, which is set at 1.65. When $EO_{AB}$ is set, the total air flow rate required by the system is calculated. The main system outputs are given in Table 3, where it can be seen that at a cell voltage of 0.60 V, the power density is 0.21 W cm$^{-2}$, which corresponds to a system efficiency of 40.1%. However, it is worth noting that this value does not consider the energy consumption of the ancillary devices such as the air blower, inverter, and HT-PEMFC cooling system.

**Table 3.** System output parameters under nominal operating conditions.

| Parameter | Value |
|-----------|-------|
| Cell voltage | 0.60 V |
| $P_{FC}$ | 0.21 W cm$^{-2}$ |
| $\dot{Q}_{FC}$ | 0.23 W cm$^{-2}$ |
| System efficiency | 0.401 |
| Afterburner exhaust temperature | 1357.62 °C |
| Ammonia specific flow rate | 0.10 g h$^{-1}$ cm$^{-2}$ |
| Air specific flow rate | 1.08 Nl h$^{-1}$ cm$^{-2}$ |
| S2 opening ratio | 0.8 |

The afterburner off gases temperature, $T_{13}$, is 1357.62 °C. The specific inlet ammonia flow rate and air flow rate calculated based on the cell active area are also given in Table 3, at 0.10 g h$^{-1}$ cm$^{-2}$ and 1.08 Nl h$^{-1}$ cm$^{-2}$, respectively. Finally, the opening ratio of splitter S2, calculated as the ratio between

the air flowrate into the cathode and the total inlet air flowrate into the system is found to be 0.8 at nominal operating conditions.

In Table 4, gas compositions of all pipes are reported. The afterburner inlet flow, AB in Table 4, is a mixture of only oxygen, nitrogen, and hydrogen. The selected composition allows to reach an afterburner adiabatic temperature of 1362.26 °C.

**Table 4.** Gas composition of all mixtures.

|  | Air (6,7,8′,8″) | NH$_3$ in (1,2,2′,2″) | NH$_3$ Ref (3,4) | Anode Out (10) | Cathode Out (9) | AB in (11,12) | AB Out (13,14,15,16) |
|---|---|---|---|---|---|---|---|
| H$_2$O | 0.0% | 0.0% | 0.0% | 0.0% | 15.5% | 0.0% | 14.5% |
| CO$_2$ | 0.0% | 0.0% | 0.0% | 0.0% | 0.0% | 0.0% | 0.0% |
| N$_2$ | 79.0% | 0.0% | 40.0% | 71.2% | 72.9% | 75.3% | 80.8% |
| H$_2$ | 0.0% | 0.0% | 60.0% | 28.8% | 0.0% | 13.5% | 0.0% |
| NH$_3$ | 0.0% | 100.0% | 0.0% | 0.0% | 0.0% | 0.0% | 0.0% |
| O$_2$ | 21.0% | 0.0% | 0.0% | 0.0% | 11.6% | 11.2% | 4.7% |

Table 5 reports the operating values of the heat exchangers and the ammonia decomposition reactor. The inlet and outlet temperatures of the hot (T$_1$) and the cold (T$_2$) gas flows are shown for the different components. Minimum temperature difference, pinch point, is also reported as $\Delta T_{min}$. Finally, logarithmic mean temperature difference (LMTD) and exchanged heat, $\dot{Q}$, are reported. All pinch points are higher than 30 °C, which is the design value of the AHE. The amount of heat exchanged in the ADR has a higher value compared to the other units since the heat has to cover also energy requirement for the chemical reaction. All data reported in Table 5 can be used for a detailed design of the heat exchangers and to calculate single efficiency of each component. However, the design of each component is beyond the aim of this study.

**Table 5.** Data of the heat exchangers.

|  |  | AHE | ADR | ABHE | CHE | CHE$_2$ |
|---|---|---|---|---|---|---|
| T$_1$ in | °C | 600.00 | 1357.62 | 592.99 | 393.00 | 218.20 |
| T$_1$ out | °C | 218.20 | 592.99 | 393.00 | 87.85 | 160.00 |
| T$_2$ in | °C | 20.00 | 570.00 | 160.36 | 20.00 | 114.02 |
| T$_2$ out | °C | 570.00 | 600.00 | 330.00 | 114.02 | 160.00 |
| $\Delta$T min | °C | 30.00 | 22.99 | 232.64 | 67.85 | 45.98 |
| LMTD | °C | 89.08 | 210.18 | 247.50 | 149.33 | 51.85 |
| $\dot{Q}$ | W cm$^{-2}$ | 0.04 | 0.11 | 0.03 | 0.04 | 0.02 |

The thermal optimization of the system is based on the three parameters $\lambda_{anode}$, T$_{12}$, and EO$_{AB}$. Anode stoichiometry defines not only the amount of hydrogen required for in the HT-PEMFC, but also the remaining amount of hydrogen that enters into the afterburner. Hydrogen reacts in the burner, providing heat to all the successive components, except the fuel cell, which produces heat. For higher values of $\lambda_{anode}$, system efficiency decreases, due to higher consumption of fuel, and the temperature of the system off gasses increases. This could be of some interest in the case of cogeneration application.

The optimization of T$_{12}$ and EO$_{AB}$ allows to distribute the heat of the afterburner among all the heat exchangers, while also maintaining the AB adiabatic flame temperature below 1370 °C. This limit was introduced to reduce NO$_x$ production according to the modeling results obtained in this work (Figure 5). In detail, higher values of EO$_{AB}$ reduce the adiabatic flame temperature but increase the consumption of air. The pre-heat temperature, T$_{12}$, is designed as a tradeoff between AB off gasses temperature and a complete heat recovery in the ABHE. Figure 6 shows the influence of AB inlet temperature on the temperature of AB off gasses and on system exhausts temperature.

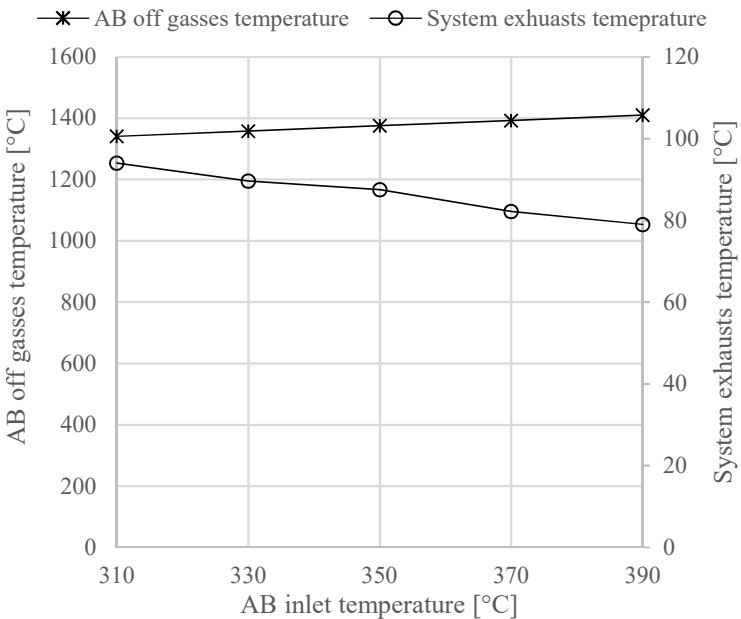

**Figure 6.** AB off gasses and system exhaust temperatures as function AB inlet temperature.

The model was also studied for different design values. The HT-PEMFC stack current density was varied from 0.25 to 0.45 A cm$^{-2}$, with step of 0.05 A cm$^{-2}$, keeping all other parameters constant. Figure 7 shows the results of the study, where the power density and system efficiency is given in Figure 7a,b shows air flow rate and $\dot{Q}_{FC}$, both as function of current density. As expected, power density increases with increasing current density, while the opposite behavior characterizes the system efficiency. This is a typical tradeoff for fuel cell systems between power density and efficiency. Both $\dot{Q}_{FC}$ and air flowrate affect the energy consumption of ancillaries.

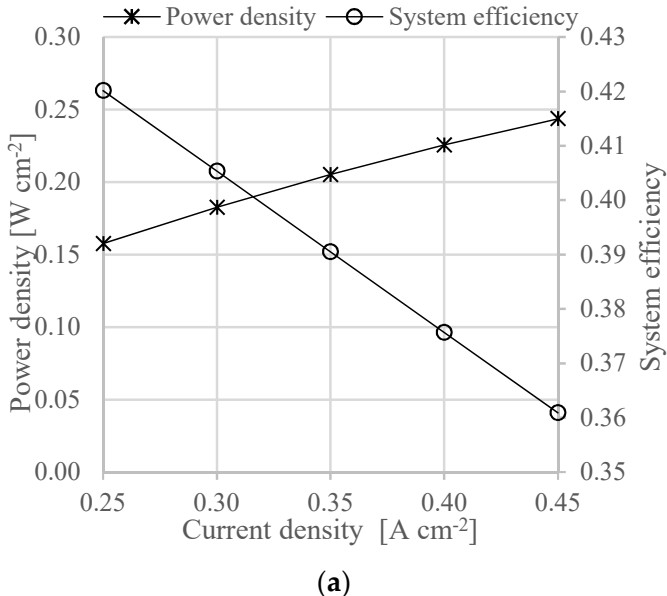

(**a**)

**Figure 7.** *Cont.*

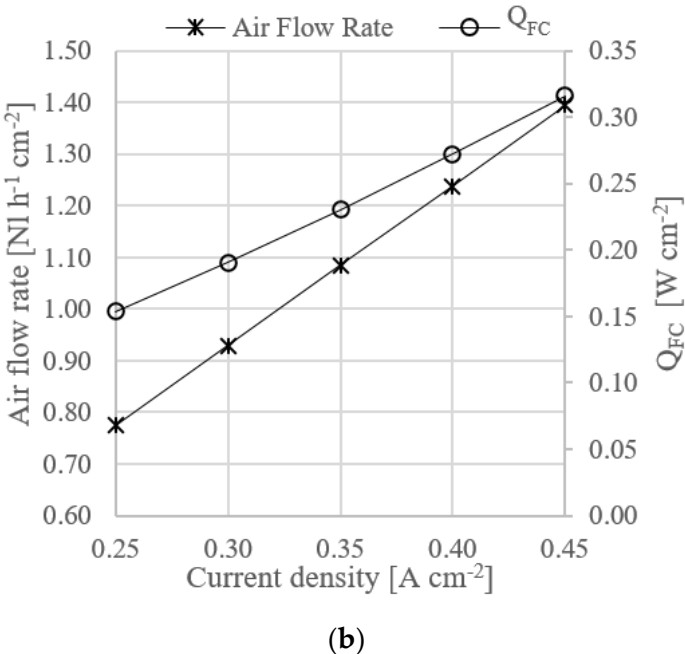

**(b)**

**Figure 7.** (**a**) Power density and system efficiency as function of current density. (**b**) Air flow rate and heat generation in the fuel cell as a function of current density.

## 4. Discussion

### 4.1. Ammonia Decomposition for the HT-PEMFC System

Ammonia decomposition experiments and models have shown that above a temperature of 600 °C, ammonia can be fully converted in the reactor over a large range of ammonia flow rates [32,34]. However, in an integrated system with an HT-PEMFC, a small amount of ammonia may still slip from the reactor outlet. In [42], it was reported that small traces of ammonia can degrade the performance of PEM fuel cells. It has also been shown experimentally that ammonia in phosphoric acid fuel cells (PAFC) can react with the electrolyte to produce $(NH_4)H_2PO_4$, which results in a more sluggish oxygen reduction reaction (ORR) at the cathode [43]. Since the commonly used PBI-based membrane electrolyte in HT-PEMFCs is doped in phosphoric acid, similar degradation mechanisms are expected to take place in the presence of ammonia in HT-PEMFCs. However, the authors also reported that the effect can be reduced by increasing the cathode potential and a complete performance recovery can be achieved when the contaminant is removed. In addition, there are several ways to reduce the ammonia concentration even further, such as absorption and adsorption processes for ammonia or membranes for hydrogen separation [44].

In comparison with methanol reforming, which is characterized by a complex system of three reactions; methanol steam reforming, methanol decomposition, and water gas shift [45], ammonia decomposition is given by the single equilibrium reaction given by Equation (4). Hence, while in the case of methanol reforming, a compromise is necessary among the different contaminants of the reformate gas depending on the reforming temperature and reactants flowrates, complete ammonia decomposition has the potential to provide clean hydrogen. This is because some of the contaminants in the methanol reformate gas are byproducts of the reforming process itself, with higher methanol conversion resulting in more CO and $CO_2$ and lower conversion ratios leaving more unconverted methanol in the reformate gas. Even though tolerances of up to 2% of CO (in the presence of $CO_2$) [46] and 3% of methanol [47] are reported in the literature, CO and methanol are nonetheless detrimental to an HT-PEMFC during long-term exposure and their combined effects could significantly lower these tolerance values. Moreover, it not possible to maintain both contaminants at low levels by increasing

the reforming temperature as this would increases the methanol conversion, thereby decreasing the methanol slip but also increasing the CO concentration in the reformate gas [45]. With ammonia decomposition on the other hand, if a proper system heat integration is done, a complete ammonia decomposition at chemical equilibrium contains only $H_2$ and $N_2$, as shown in Figure 3, which does not have any degrading effects on the fuel cell stack performance as can be seen from the experimental results in Figure 4.

*4.2. Ammonia Fueled HT-PEMFC System Analysis*

Based on the results in nominal condition, it is possible to design a 10 kW power system based on ammonia as a fuel and HT-PEMFC. Main system parameters are reported in Table 6. For a 10 kW system, around 295 cells of 165 $cm^2$ of active area are necessary. Ammonia inlet flow rate of 1.37 g $s^{-1}$ is required as fuel consumption and an air flow rate of 14.68 Nl $s^{-1}$ has to be provided by the air blower.

Table 6. Data of the 10 kW system.

| System Power | 10.00 | kW |
|---|---|---|
| Number of Cells (165 $cm^2$) | ≈295 | |
| $NH_3$ Flow Rate | 1.37 | g $s^{-1}$ |
| Air Flow Rate | 14.68 | Nl $s^{-1}$ |

It is important to underline that the detailed design of the heat exchanger is necessary to evaluate the efficiency of the components. The ammonia decomposition reactor has to be designed to evaluate the amount of catalyst required to reach the complete decomposition of the ammonia. The natural evolution of the study is the evaluation of HT-PEMFC tolerance to $NH_3$ traces. The limit of the technology could bring two different scenarios. If the tolerance is extremely low, an additional clean-up unit may be necessary, increasing the complexity of the system. If the tolerance is in the order of percentage, it is possible to reduce ADT temperature moving all the system temperature to lower values. The latter scenario is interesting as it brings the $NH_3$ system closer to the existing methanol-fueled HT-PEMFC power units and it will be possible to recover part of the know-how included and already developed components for the balance of the plant. In addition, the reduction of ammonia decomposition temperature reduces the dimension of the heat exchangers due to the general reduction of LMTD values. In the definition of the off-design operation and start-up, the direct use of ammonia in the afterburner has to be investigated. In particular, the operation of the afterburner with a pure ammonia mixture requires a deeper investigation.

## 5. Conclusions

In this work, a design concept of a heat-integrated ammonia-fueled HT-PEMFC system is presented. Zero-dimensional models of the reactors and heat exchangers were developed, including those of the afterburner and the ammonia decomposition reactor. The study can be considered as a first attempt to demonstrate the thermodynamic feasibility of such a system. According to the chemical equilibrium studies in this work, the ammonia decomposition reactor can fully convert the ammonia into hydrogen and nitrogen. However, the process is endothermic and high operating temperatures of above 600 °C need to be ensured in the decomposition reactor, which in the proposed concept, is provided by the afterburner.

Moreover, the $NO_x$ emission of the combustion processes involved in the proposed system have been analytically investigated. The model showed that both ammonia/air combustion and $H_2$/air combustion results in a limited amount of NO production of around 22 ppm and 4 ppm, respectively, below 1370 °C. Therefore, the afterburner temperature should be kept below this temperature in order to limit the production of NO during the combustion process. Since the combustion of ammonia is characterized by low flame speed and long ignition time, in the proposed concept, ammonia combustion is only used during the startup phase, and during normal operating conditions, the unreacted hydrogen

at the anode outlet is used in the afterburner as it can sustain the system thermal requirements with a fuel cell anode stoichiometric ratio of 1.37.

An experimental test on an HT-PEMFC short stack was also conducted to observe the cell electrical performance using a fuel mixture with a composition of hydrogen and nitrogen gas similar to the one at the outlet of the ammonia decomposition reactor. The experimental results showed that in the absence of the $NH_3$ slip in the feed gas (via complete decomposition or gas purification), a hydrogen–nitrogen gas mixture at concentrations corresponding to a typical ammonia decomposition process does not degrade the fuel cell stack performance.

At system level, the proposed design allows to reach a total efficiency of 40.1% at a power density of 0.21 W cm$^{-2}$. The heat recovery strategy allows to feed the ammonia decomposition reactor and to preheat the gas before entering the HT-PEMFC.

**Author Contributions:** Conceptualization, G.C., V.L., S.L.S. and S.S.A.; methodology, G.C., V.L., S.L.S. and S.S.A.; software, G.C., V.L. and S.S.A.; validation, G.C., V.L. and S.S.A.; formal analysis, G.C., V.L. and S.S.A.; investigation, G.C., V.L., S.L.S. and S.S.A.; resources, G.C., V.L. and S.S.A.; data curation, G.C., V.L. and S.S.A.; writing—original draft preparation, G.C., V.L. and S.S.A.; writing—review and editing, G.C., V.L., S.L.S. and S.S.A.; visualization, G.C., V.L. and S.S.A.; project administration, G.C. and S.S.A.; funding acquisition, G.C. and S.S.A. All authors have read and agreed to the published version of the manuscript.

**Funding:** The activities were supported, in the frame of a scientific collaboration, by ENVIU, PROTON VENTURES BV and C-JOB & PARTNERS BV, which are acknowledged. This research has also received funding from the Danish Energy Technology Development and Demonstration Program (EUDP) through the projects COBRA Drive (grant number—64018-0118). The study was supported by the Fuel Cell and Hydrogen 2 Joint Undertaking under grant agreement No. 736648 (NET-Tools project).

**Conflicts of Interest:** The authors declare no conflict of interest. The funders had no role in the design of the study; in the collection, analyses, or interpretation of data; in the writing of the manuscript, or in the decision to publish the results.

## Abbreviations

The following abbreviations and symbols are used in this manuscript:

| | |
|---|---|
| AB | Afterburner |
| ABHE | Afterburner Heat Exchanger |
| ADR | Ammonia Decomposition reactor |
| ADT | Ammonia Decomposition temperature |
| AHE | Ammonia Heat Exchanger |
| CHE | Cathode Heat Exchanger |
| EO | Excess of Oxygen |
| LHV | Low Heating Value |
| LMTD | Logarithmic Mean Temperature Difference |
| HT-PEMFC | High Temperature Proton Exchange Membrane Fuel Cell |

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
