# Peer review of "System Design and Modeling of a High Temperature PEM Fuel Cell Operated with Ammonia as a Fuel"

_energies, doi:10.3390/en13184689_

Round 1

Reviewer 1 Report

The Authors demonstrate the concept of a HT-PEMFC system that is fed by hydrogen generated from ammonia. According to their calculations (supported by some experimental results) the FC can operate with H2 + N2 gas mixture at an efficiency of 44.2% based on ammonia at 0.4 A/cm2 of current density and 0.66 V of voltage, which is convincing enough that the topic is worthy of attention. The main question is the possibility of coupling an NH3 decomposition reactor to an HT-PEMFC, since residual NH3 can have detrimental effect on PEMFC parts. So far, SOFCs were successfully operated with NH3, but PEM technology would be more appropriate in many applications. I think the Authors give a promising answer to this question by their model, that is interesting and it is a step forward in this area. The concept and its presentation are clear, the text is well written (some typos have to be corrected in the proof reading process), easy to read, and the manuscript will attract attention, therefore it can be published after some minor corrections I suggest below.

Some suggestions for minor revisions:

    The abstract says: "This study presents, for the first time in literature, an integration study of ammonia as a fuel and a high temperature polymer electrolyte membrane fuel cell (HT-PEMFC) as an energy conversion device."
    In order to distinguish this concept from direct ammonia FCs, I suggest to define ammonia as hydrogen carrier, because it is first transformed to H2+N2 and the FC unit operates as a normal HFC converting H2+O2 to H2O in the system by the Authors. In the introduction the Authors themselves make this distinction (line 76).
    Some references might by also included:
    https://doi.org/10.1016/j.rser.2016.01.120
    Line 38: "thereby increasing energy density" Please add "volumetric" Ethanol = ethanol
    Line 66: 75% mol = 75 mol%
    Line 128: "HT-PEMFC stack was tested with pure hydrogen and with a mixture of 68.3% hydrogen and 31.7% nitrogen in the anode feed. Each gas composition was run for 24 hours at 0.4 A cm-2, where the stoichiometric ratios were set to 1.3 and 2.5 for anode and cathode, respectively."
    This explanation is a little confusing to me. Please clarify the exact conditions.
    Lines 160, 169, 175, 211, 302, 324 and 329: Please check the text as some fragments/links appear unfitting.
    Line 232: "...where Ru-based catalyst have shown the best results. Other catalysts include nickel, platinum and iridium, all characterized by a high cost."  Nickel expensive vs. Ruthenium? I doubt.
    Line 239: I disagree with the use of equation (5). Instead of concentration write molar fraction that is unitless (x), otherwise 1 - c can adopt a wide range of invalid values.
    Lines 336-337: these lines seem to be fragments.
    Table 3: what do the numbers refer to? (Figure 1?)
    Line 368: "...from 0.3 to 0.5 A cm-2, with step of 0.1 A cm-2" I see 0.05 A/cm2 steps.

Reviewer 2 Report

I recommend to accept 

Author Response

We thank the reviewer for the positive feedback.

Reviewer 3 Report

Energies

Manuscript Number: 892077

Title: System design and modelling of a high temperature PEM 2 fuel cell operated with ammonia as a fuel

Authors: Giovanni Cinti, Vincenzo Liso, Simon Lennart Sahlin and Samuel Simon Araya

Summary

The paper presents studies of system design and modeling of HT-PEMFC operated with NH3 as fuel. The authors presented and discussed a design concept of a system which integrates NH3 decomposition reactor and HT-PEMFC. In this concept, NH3 decomposes to H2 and N2 and then the mixture goes to HT-PEMFC stack. The authors provided zero-dimensional models of the full process of H2 production from NH3 including ammonia decomposition reactor, heat exchangers, afterburner etc. According to the modelling results and analysis of chemical equilibrium studies the ammonia decomposition reactor can fully convert the ammonia into hydrogen and nitrogen at high operating temperatures of above 600C. The authors also evaluated NOx emission involved in the proposed design. The model showed that both ammonia/air combustion and H2/air combustion results 434 in a limited amount of NO production of around 22 ppm and 4 ppm, respectively, at the low operating temperatures below 1370C. Experimental work was done with HT-PEMFC short stack to demonstrate performance of using diluted H2 anode feed gas. However, it was assumed that there is no any NH3 contamination of the fuel gas stream. The authors stated that the concept design reached total efficiency of 40%.

The submission is important to one of the goals of fuel cell field of diversification of H2 sources and elimination of CO2 emission. The work also partially serves as a reference work combining research in NH3 decomposition and HT-PEMFC performance. The manuscript is very well written and presents results and discussion in detailed and thorough manner.

I noticed several minor misprints:

  1. Abstract: “LHV” abbreviation was used in the abstract, but I could not find what it stands for.
  2. Page 3, line 102. “…HT-PEMFCs have been extensively studies in conjunction…” I assume it should be “…HT-PEMFCs have been extensively studied in conjunction…”
